# Cherenkov Radiation in Optical Fibres as a Versatile Machine Protection System in Particle Accelerators

**DOI:** 10.3390/s23042248

**Published:** 2023-02-16

**Authors:** Joseph Wolfenden, Alexandra S. Alexandrova, Frank Jackson, Storm Mathisen, Geoffrey Morris, Thomas H. Pacey, Narender Kumar, Monika Yadav, Angus Jones, Carsten P. Welsch

**Affiliations:** 1Department of Physics, School of Physical Sciences, Faculty of Science and Engineering, University of Liverpool, Liverpool L69 7ZE, UK; 2Cockcroft Institute, Sci-Tech Daresbury, Daresbury WA4 4AD, UK; 3ASTeC/STFC, Sci-Tech Daresbury, Daresbury WA4 4AD, UK

**Keywords:** optical fibre, beam loss, RF breakdown, accelerator physics

## Abstract

Machine protection systems in high power particle accelerators are crucial. They can detect, prevent, and respond to events which would otherwise cause damage and significant downtime to accelerator infrastructure. Current systems are often resource heavy and operationally expensive, reacting after an event has begun to cause damage; this leads to facilities only covering certain operational modes and setting lower limits on machine performance. Presented here is a new type of machine protection system based upon optical fibres, which would be complementary to existing systems, elevating existing performance. These fibres are laid along an accelerator beam line in lengths of ∼100 m, providing continuous coverage over this distance. When relativistic particles pass through these fibres, they generate Cherenkov radiation in the optical spectrum. This radiation propagates in both directions along the fibre and can be detected at both ends. A calibration based technique allows the location of the Cherenkov radiation source to be pinpointed to within 0.5 m with a resolution of 1 m. This measurement mechanism, from a single device, has multiple applications within an accelerator facility. These include beam loss location monitoring, RF breakdown prediction, and quench prevention. Detailed here are the application processes and results from measurements, which provide proof of concept for this device for both beam loss monitoring and RF breakdown detection. Furthermore, highlighted are the current challenges for future innovation.

## 1. Introduction

Modern particle accelerators often store extreme energy densities within a particle beam [1,2,3]. Anomalous behaviour of this beam can cause issues such as excessive radioactive activation to beamline components or areas, or even damage to accelerator components. Both of these effects produce additional strain on resources with increased maintenance, hardware turnover, and facility downtime. Machine protection systems have been developed to counteract these issues and alleviate the strain on these resources. Instrumentation is capable of detecting aberrant behaviour along the beamline on the microsecond scale [4,5]. This rapid detection allows operators, feedback systems, or other automated processes to rectify beam behaviour either by tuning machine parameters or, in extreme cases, dumping the particle beam to prevent further damage or activation.

For the case of beam loss monitors (BLMs), a range of instrumentation has been developed in the field, each with a specific application in mind. By far, the most common is the ionisation chamber (IC) [6,7]. These devices are typically ∼50 cm in length and are positioned at discrete locations along a beam line. Each provides an independent absolute loss value at a single location, which can be calibrated to produce an absolute dose. This dose value is then used to assess component or area activation or to plan maintenance and long term hardware replacements. On a much shorter time frame, ICs can be used to monitor instantaneous lose intensities; this information is critical to limiting machine damage in the event of a significant beam loss incident. Machine protection systems directly monitor this output against a predetermined threshold, above which the beam is dumped and the machine is reset. The main drawback to this method is the discrete nature of the data collected. The ICs are placed at fixed locations and only provide loss information at that location. This means that the loss location resolution of such a system is inherently linked to the number of sensors in place on a beamline; for even medium-sized accelerator infrastructures, sub-metre location resolution can be extremely resource heavy. This can lead to prioritisation of certain sectors of beamline for BLM installation, which in turn leads to sections with >10 m of unobserved beamline and potential beam loss. Besides ICs, there are pCVD, SEMs, PMTs, and many more [5]. All of these devices have the same characteristic; absolute dose measurement at independent discrete locations. They are an incomplete, disconnected series of data points along a beamline.

Another facet of machine protection is radio frequency (RF) breakdown detection and mitigation. Standard acceleration technology utilises a high conductivity metal cavity to accelerate the charged particles in the machine; for normal conducting this is typically copper, for superconducting this is typically niobium. RF fields are generated and propagated into these cavities, where the correct particle beam phase can produce acceleration from the standing waves created inside the cavity. These accelerating field gradients can be ∼100 MV/m or more. A limiting factor of the accelerating gradient is the material of the cavity itself and the process of RF breakdown. An exact theory of RF breakdown is still to be agreed upon, but the general consensus is that the process proceeds as follows [8,9]. With the application of an RF field, the interior surface of the cavity can have electrons pulled free, known as electron field emission. These electrons are accelerated by the RF fields within the cavity and can strike the cavity surface once more, generating further electrons. This field emission is also accompanied by gas desorption from the cavity surface. This gas is ionised by the accelerated electrons, forming a plasma, which creates an electric short to the RF power, causing the input power to be reflected back to the source. The arc created by this short can also cause significant damage to the cavity inner surface. The individual steps of this process can easily cascade, destroying the accelerating field within the cavity, and damaging the cavity surface. This is known as an RF breakdown. It is extremely difficult to predict the onset of such events, but they are made worse by machining/fabrication procedures or by surface defects, such as those caused by RF breakdowns. This can lead to a negative feedback loop, where RF breakdowns cause further RF breakdowns at lower field strengths, ruining an accelerators performance. It is therefore of utmost importance to intervene and mitigate a breakdown event as soon as possible, to minimise any long-term impact. Diagnostics are therefore designed to react as quickly as possible. One such system utilises the actual RF input pulses. A correctly tuned cavity should accept RF pulses with a high efficiency. However, when a breakdown event occurs, the reflected power is significantly increased. Monitoring this reflected power allows operators or automated systems to modulate the RF power into a cavity and prevent a breakdown from causing significant damage. Another system in common use monitors the vacuum pressure within a cavity, as this pressure increases during a breakdown event via the desorption process mentioned above. This output is also used to moderate the RF fields within the cavity. A distinct disadvantage which is inherent to these diagnostics is the reactive operational model. The systems require field emission, or even a breakdown event, to occur at a significant level before action can be taken. Efforts have been taken to utilise novel machine learning techniques in the prediction of RF breakdown ahead of time, but success has been limited due to the data that are available with these techniques [10,11].

The final major component of a machine protection system is quench detection. In facilities operating superconducting elements, monitoring this superconductivity is a prerequisite to ensuring these elements are not irreparably damaged. A quench occurs when the temperature of a superconducting element rises above its critical temperature, when it loses its superconducting properties and begins to demonstrate electrical resistance. For example, with a superconducting electromagnet carrying an extremely high electrical current, this sudden onset of resistance could be very dangerous, generating a significant heat load. Common causes of quench are in fact beam loss and RF breakdown. Therefore, in principle, the same diagnostics as listed above could be used. However, practically this is not always possible, due to the large cryostats used to cool the superconducting beam line elements. Other diagnostic systems are used to directly monitor the resistance of the superconducting material and the temperature of the cooling media [12,13,14].

The three areas outlined so far, beam loss, RF breakdown, and quench detection, are the main concerns of most machine protection systems. A common theme can be seen across all the instrumentation highlighted above: each system is reactive, with no predictive capability; the devices are independent of one another, with little utilisation of the possible data links and patterns between devices; and achieving full facility coverage can be expense, time consuming, and require high levels of long term maintenance.

To this end, a new machine protection system based on optical fibre technology has been developed. This system could run standalone, or be used to augment and enhance conventional systems for more effective machine protection functionality. The sensing medium in this device is the optical fibre itself. Energetic charged particles passing through the fibre generate a pulse of Cherenkov radiation [15,16,17,18]. This is broadband radiation emitted by a charged particle when it passes through a medium with a speed larger than the phase velocity of light in that medium [19]. Once the material properties are accounted for, silica fibres in this case, the radiation peaks in the optical region of the electromagnetic spectrum [20]. The light is generated, captured, and propagated by the optical fibre. Time of flight measurements at either end of the optical fibre allow the precise location of the source of the Cherenkov pulse to be measured. This process will be discussed further in the next section. This is the first instance of such a device being used to measure two different machine protection-related phenomena, with no modifications required to switch between the two operational modes.

The optical fibre is a sensor rather than, for example, a relatively compact IC. Therefore, as a BLM, large sections of beamline can be covered, providing continuous monitoring and no dark spots where losses can be missed. With the appropriate calibration, as with ICs, absolute loss/dose information can also be achieved. In the realm of RF breakdown, the optical fibre can be used to measure the loss showers of those initial electrons pulled from a cavity surface by field emission and the subsequent build-up. This provides an inherent sensitivity to the onset of a breakdown, rather than detecting when a breakdown has occurred. Furthermore, as the output is continuous and the measurement instantaneous, the only significant time delay in the system is the length of the optical fibre. With a dedicated, small-scale system, rapid and high sensitivity breakdown measurements can occur. This build-up tracking-type output is also suited to the application of machine learning techniques for the prediction of RF breakdown events ahead of their arrival. This would lead to earlier, smaller mitigation actions for accelerator operators, increasing machine up-time, output, and overall efficacy. These specific applications of the system, along with machine learning integration, are discussed further in the following sections.

## 2. Materials and Methods

### 2.1. Detection Mechanism

In the majority of applications under review, the Cherenkov sensor will detect secondary particles rather than primaries; that is, the charged particle shower generated by an energetic particle striking a surface. This burst of charged particles, typically electrons, is generated by some external event, this could be beam loss or RF breakdown. The charge shower propagates in free space to the optical fibre where some small solid angle will intersect the fibre. This solid angle is dependent on the diameter of the fibre core. Wider fibres provide a larger signal as a larger number of particles in the shower interact with the fibre. However, this larger solid angle also increases the pulse width of the Cherenkov radiation and limits the resolution of the device [16,17]. Tests regarding the effect of the fibre core and cladding size have been conducted in previous work, and a range of 200–600 μm was found to be optimal depending on the application. For example, a lower energy beam would create less charged particles in a loss event than a higher energy beam, meaning a larger diameter fibre would be required to collect sufficient signal for analysis.

Once the particles make it to the fibre, the detection process begins. A schematic of the generation, capture, and propagation of the Cherenkov radiation within the optical fibre is presented in Figure 1. The charged particles pass through the fibre relatively unscathed. As stated earlier, those particles whose velocity is larger than the phase velocity of light within the fibre will generate Cherenkov radiation. This velocity can be calculated from relativity and optical theory. Given that the phase velocity of light in a medium is defined as:(1)$$v_{p}=\frac{c}{n},$$
where $v_{p}$ is the phase velocity of light in a medium with a refractive index of *n* and *c* is the speed of light in vacuum, for silica optical fibres (*n* = 1.46), this gives $v_{p}=0.69c$. As β=v/c, with *v* as the particle speed, this means for β>0.69, Cherenkov radiation will be generated. Relativistic theory defines
(2)$$KE=m_{0}c^{2}\left(\frac{1}{\sqrt{1-\beta^{2}}}-1\right),$$
where *KE* is the kinetic energy of the particle and $m_{0}$ is the rest mass of the particle. Combining Equations (Equation 1) and (Equation 2), and continuing with the example of an electron in silica, gives *KE* = 186 keV. Therefore, any electron passing through the fibre with KE > 186 keV will produce Cherenkov radiation within the fibre.

This value is of course an estimate, as over a broad enough spectrum, most (if not all) materials are dispersive, i.e., $n\equiv n\left(\lambda \right)$. As most materials have n<1 at λ < 100 nm, this also means that Cherenkov radiation cannot be produced, as $v_{p}>c$ at such low wavelengths. The number of Cherenkov photons per unit length per unit wavelength can be calculated using the work of Frank and Tamm [19]:(3)$$\frac{\delta^{2}N}{\delta x\delta \lambda }=2\pi \alpha \frac{Z^{2}}{\lambda^{2}}\left(1-\frac{1}{\beta^{2}n^{2}\left(\lambda \right)}\right),$$
where *x* is a distance travelled by the source particle, λ is the wavelength of the emitted photons, *Z* is the particle unit charge (i.e., for an electron this would be −1), and α is the fine structure constant $\left(\approx 1/137\right)$. From Equation (Equation 3), it can be shown that for a given particle energy, the number of Cherenkov photons reduces with increasing wavelength. Therefore, the Cherenkov generation peaks in the UV and drops when moving to the optical and infrared.

This behaviour is actually opposite to the transmission behaviour found in optical fibres. For the diameter range of interest, attenuation with distance is at a minimum in the infrared [20]. This then increases as the wavelengths drops to the optical and then to the UV, where most radiation is lost. These two effects combined lead to the actual Cherenkov radiation signal in the optical fibre of this system peaking in the optical region.

The photons produced in the fibre, as per Figure 1, are produced in a forward facing cone. The opening angle of this cone is defined as:(4)$$\theta_{c}=\arccos\frac{1}{\beta n\left(\lambda \right)},$$
where $\theta_{c}$ is the opening angle, as shown in Figure 1. With n=1.46, the estimate for silica at optical wavelengths used above, $\theta_{c}$ = 47${}^{\circ }$ for β≈1 from Equation (Equation 4). For lower energy particles, the cone will be wider. Once the photons are within the fibre, those whose direction of travel falls within the capture angle for total internal reflection, defined by the fibre cladding and core, are captured and propagated down the fibre. The large spread in the Cherenkov cone leads to photons being distributed between downstream and upstream propagation directions. This distribution depends on the incoming direction of travel and energy of the source particle. For example, in beam loss, the charge shower is typically directed more towards the direction of travel on the main beam; this leads to a large number of photons travelling downstream and less upstream. This uneven distribution can be advantageous as the downstream detector receives an intense signal, whereas the upstream detector receives a very narrow pulse; this provides a balance between resolution and signal-to-noise.

Detectors are placed at the ends of the fibres to measure the longitudinal properties of the Cherenkov pulse. The photo-sensor used in this device is a silicon photo-multiplier (SiPM) [21], the Hamamatsu Photonics multi-pixel photon counter (MPPC, specifically the S12572-010C) [22]; these are an array of avalanche photodiodes operating in Geiger mode. SiPMs are rarely used in the machine protection sector, where common sensors are ICs and photomultiplier tubes (PMTs) [23]. The major benefits of using SiPMs include low voltage, high gain operation; superb timing properties; high sensitivity; and immunity to magnetic fields [21]. Each of these factors contribute to features of the system, but the standout characteristic is the timing performance. The system developed in this work is capable of resolving pulses in the range of ∼10 ns. The sensor is typically paired with a high bandwidth (∼1 GHz) amplifier, in this instance, a transimpedance amplifier, to provide gain whilst maintaining the temporal properties of the light pulse.

The installation process is often application dependent. Generally, the method used involves placing the optical fibre in close proximity to the beamline. Practicalities have to be observed when doing this, as demonstrated in Figure 2. In theory, the optical fibre as a sensing device allows a user to run the fibre through machine elements (e.g., quadrupoles, dipoles, etc.) and reduce the distance to the beamline to a minimum, which would improve the resolution of measurements. This is because a narrower section of any charge shower would be measured. Practically, however, this approach can run into issues. Accelerator components often require maintenance and regular access; if the fibre is passing through a component which needs attention, the fibre would have to be removed. This could lead to a large amount of fibre being extracted from the beamline on a semi-regular basis. Optical fibres are delicate components, and regular unnecessary handling would lead to damage, which likely would not be noticed until the machine was running again and the area was no longer accessible. A more utilitarian approach is to sacrifice a small amount of resolution for long term fibre stability, by placing fibres where they can be left indefinitely. The distance away from the beamline which is acceptable is beam dependent, with larger energy beams allowing larger distances.

Once installed, the optical fibres can be used to route the signal out of the radiation controlled area. This maximises the speed with which the signal can be analysed and reduces the maintenance burden of the device, as no electronics are required within the radiation area.

### 2.2. Application to Beam Loss

The application of detecting Cherenkov radiation via optical fibres for beam loss monitoring purposes has already been demonstrated with varying levels of success [16,18,20,24]. The focus of these particular measurements was to demonstrate a much improved loss location resolution, whilst also defining a new practical loss location resolution. Two methods of operation are possible with beam loss monitoring. The first, as briefly mentioned earlier, uses a detector at either end of the optical fibre. By synchronising the detectors with the master clock of a facility and utilising the upstream/downstream split of the Cherenkov radiation, a time of flight measurement can be made. An example of how this measurement system works is presents in Figure 3.

An initial thought would be the requirement for a measurement trigger; something to time against. A master clock normally provides a trigger signal for every instance of a particle bunch generated within an accelerator, and is a stable source to measure against. This bunch will travel to a certain location and produce a loss, which will in turn produce upstream and downstream Cherenkov pulses, as detailed above. The time measured on the two detectors will be:(5)$$\tau_{u,d}=\frac{x_{acc}}{v_{part}}+\frac{x_{u,d}n}{c},$$
where $\tau_{u,d}$ is the time measured on the upstream and downstream detectors, $x_{acc}$ is the distance the particle will have travelled from the clock trigger point to the loss location, $v_{part}$ is the particle velocity (typically ∼c), and $x_{u,d}$ is the distance the Cherenkov signal has travelled from the loss location to the upstream and downstream detectors. Two constants can be used to find the loss location. The first is the length of the fibre, $L_{f}$, which is a known quantity defined as $\tau_{f}c/n$, where $\tau_{f}$ is the time the signal would take to traverse the entire fibre. The second, $x_{acc}/v_{part}$, is fixed. Using these constants and Equation (Equation 5),
(6)$$\Delta \tau =\tau_{u}-\tau_{d}=\frac{x_{u}n}{c}-\frac{x_{d}n}{c},$$
furthermore, therefore,
(7)$$\tau_{f}-\Delta \tau =\left(\frac{x_{u}n}{c}+\frac{x_{d}n}{c}\right)-\left(\frac{x_{u}n}{c}-\frac{x_{d}n}{c}\right)=\frac{2x_{d}n}{c}.$$

From Equation (Equation 7), the loss position can be extracted directly. It is now also obvious that a trigger signal is not required at all. The time of arrival difference between the two detectors is equivalent to Equation (Equation 6); all that is required is a means of synchronising the detectors. The loss location, as measured from the downstream end of the optical fibre, is defined as:(8)$$x_{d}=\frac{c}{n}\frac{\tau_{f}-\Delta \tau }{2}.$$

The second method for detecting the loss location uses only one detector, but requires a trigger signal. The benefit of this method is the cheaper, simpler installation and the simpler synchronisation requirements. This method uses known, controllable loss mechanisms to generate a calibration from time to distance. These loss mechanisms are typically screens or dipoles. With several measurement points, a straight line fit can be created which accounts for $x_{acc}$ and provides a resolution comparable with the time of flight setup. The main drawback of this system is the time needed to perform the calibration measurements themselves. However, this method can avoid complications in large systems where synchronisation over >100 m would be difficult, or where the installation of a second sensor is not practical.

### 2.3. Application to RF Breakdown

The application of this technology to RF breakdown detection is still an area of active research [25,26]. A variety of methods and measurements are possible and the optimal solution is yet to be found. The most obvious application is in RF breakdown detection in a chain of accelerating structures. This would be analogous to the BLM application in that the signal would allow users to pinpoint which cavity in the chain is causing issues. Often, this information can require intense analysis of all the cavity traces to identify the suspect cavity [13,27]; the optical fibre could tell operators this information immediately. This is demonstrated in the schematic in Figure 4. This would be equally applicable in other RF structures such as RF waveguides.

A variation of this would be to focus on a single cavity. Through thoughtful placement of the fibres, the exact location of hot spots during a breakdown could be identified. The success of this method is dependent upon how the size and shape of the cavity compares to the location resolution of the optical fibre system. By shortening the fibres to a single cavity, this would also shorten the time delay between the build up of a breakdown and the detection. In principle, this could be quicker than existing diagnostics, which operate at the ∼10 μs level. Continuous or integration measurements can be used [25,26]; the former would give instantaneous field emission information (e.g., location, intensity, and duration), the latter would give a measure of the charge generated in a given time frame.

As discussed above, with beam loss and RF breakdown being the main causes of quench in superconducting beamline elements [28,29], any device aiming to monitor, control, and reduce these events will directly contribute to quench prevention. As such, there are currently no plans for a dedicated quench detection version of this device. Some tests using liquid nitrogen to cool the fibres have been carried out, which showed that, in principle, the optical fibre could be placed inside of a cyrostat. The necessity of this is still to be established.

## 3. Results

The following section will provide an overview of several key benchmarking results conducted as part of an installation at the Compact Linear Accelerator for Research and Applications (CLARA) facility (STFC, UK). Measurements of beam loss and RF breakdown were conducted. Several initial test measurements with a new system will also be presented, which significantly improve upon the sensitivity and resolution of the existing setup. At the time of the measurement programme, CLARA was a 50 MeV electron beam linear accelerator, operating at 10 Hz, with bunch charges of up to 250 pC. Four optical fibres were installed on the front end of the facility; this was a 10 m section starting from the photocathode electron gun [30]. A schematic of the beamline and the optical fibre installation is presented in Figure 5.

In this instance, due to the relatively short fibres and the short length of time that the fibres would be installed, the decision was made to thread the fibre through machine elements, maintaining close contact with the beam line at all times. The resulting installation in demonstrated in Figure 6. These images demonstrate how flexible the system is and how close to the beam the optical fibres can be positioned. The fibres are completely insensitive to magnetic fields and X-ray background; therefore, it is easy to envision this sort of installation inside delicate devices such as undulators. As shown in Figure 5, this installation allowed BLM capabilities to be tested using screens, collimators, and magnets, whilst also passing a 2.5 m long S-band (2998.5 MHz) accelerating copper cavity [26], which provided a means of testing RF breakdown detection.

The system installed utilised a single detector, calibration-based, as described above in Section 2.2. This enabled four fibres to be placed surrounding the beam pipe (above, below, left, and right) using four detector units, rather than two with a detector on either end of the fibres. In addition to this, two fibre diameters were used to provide a working comparison: two 400 μm core and two 600 μm core. Both types are a high hydroxyl (OH), Technology Enhanced Clad Silica (TECS) multimode fibre with 0.39 NA. These were split into an opposing pair on the horizontal axis and an opposing pair on the vertical axis. As this was a linear accelerator, the losses in the horizontal axis were expected to be symmetrical, as were those on the vertical axis; this would allow a comparison in performance between the two core diameters for different measurement scenarios. The calibration procedure was conducted using five predetermined points of loss, as described above, and shown in Figure 5.

### 3.1. Beam Loss Monitoring

A series of tests were conducted to benchmark the systems performance. The first was an assessment of the Cherenkov signal linearity with bunch charge. The initial bunch charge for these tests was 100 pC; this was gradually degraded using collimation methods, whilst the bunch charge was monitored using a Faraday cup at the beam dump. By inserting a beam imaging screen downstream of the collimation, the beam loss signal could be monitored as a function of bunch charge. An example of a series of measurements for a single fibre is presented in Figure 7. A sharp peak, along with a ringing effect, can be seen for each bunch charge value. The ringing in the tail of the distribution is attributed to electrical noise in this version of the custom transimpedance amplifier used in the detector units.

The signals for each of the four fibres are presented in Figure 8. Aside from the left fibre, where the signal was so intense it saturated the digitisation system, the first thing to note is the linear response of all four fibres. Second, is that the 600 μm core optical fibres did indeed see a larger signal than their counterpart 400 μm optical fibres, to the detriment of the left fibre in this instance. It is also apparent that the vertical axis saw a significantly less intense shower than the horizontal axis. Beam physics simulations seem to support this, but it cannot be ruled out that the fibre placement relative to the beam pipe may have played a factor. This comparison of axes was not an initial goal of the measurements, so the system was not setup to definitively assess this effect. A final worthwhile result was that measurements from losses of a 10 pC bunch have been demonstrated, showcasing the overall sensitivity of the system.

Assessing the loss of location accuracy and resolution was the next goal of the measurement programme. There are two ways to define resolution: how well the peak of a single loss point can be defined and how clearly multiple loss points can be resolved. The former definition is quite limiting. From Figure 7, the peak of this distribution is easily found, with any simple curve fitting analysis placing the resolution on the ∼1 cm scale. However, losses in an accelerator beamline are very rarely singular; therefore, this definition of location resolution is at best unhelpful and at worst misleading. Hence, it was the latter definition of loss location resolution that was measured here. To achieve these multiple loss scenarios, starting upstream immediately following the RF cavity, screens were lowered into the beam until a loss was registered on the system, whilst not completely intercepting the beam. This was carried out for each screen one-by-one, moving downstream. This method enabled multiple loss location scenarios to be observed. Two examples of these measurements are presented in Figure 9 and Figure 10.

Figure 9 is an example of a three loss scenario for two of the four fibres. The top fibre produces less signal due to the fibre layout, despite the larger core, this was because in most instances the fibres on the horizontal axis could be placed closer to the beamline. Figure 10 is an example of a four loss scenario for the right fibre shown in Figure 9. It is clear to see that the position of the individual loss peaks from the previous plot are maintained with the introduction of a new loss. The relative intensity of the peaks however does vary, as different amounts of charge are lost with the introduction of additional barriers to the beam path. The overall scale on each graph varies slightly as attenuation was introduced to remove the saturation found in the measurements above. Even with the electronic ringing discussed earlier, the individual loss peaks are easily distinguishable. This figure also includes a fit curve for a four Gaussian estimation of the signal; this allows the individual loss peaks to be separated and the individual loss characteristics to be interrogated. The average 1σ width of these Gaussians is 1 m, which leads to a theoretical measure of the loss location resolution. However, the accuracy of the loss location needed to be assessed with a more practical method; the accuracy being how the loss locations from the fibres related to the actual loss locations on the beamline. From these figures, the loss location accuracy of the system was estimated at approximately 0.5 m, i.e., individual losses from the fibres were on average accurate to within 0.5 m of the expected loss location and the distances between losses in the multiple loss scenarios were accurate to within 0.5 m. This accuracy and resolution is one or two orders of magnitude better than facilities that rely on discretely placed ICs. Most accelerator beamlines have components spaced at larger than 0.5 m, which means this device could inform an operator which beamline element specifically is causing the beam loss.

### 3.2. RF Breakdown

RF breakdown measurements were conducted with exactly the same system as the beam loss measurements, no modifications were made. This measurement programme was conducted as part of the unloaded RF conditioning; no beam was present in the facility, just RF power being delivered from the klystrons to the accelerating cavity. Two conventional diagnostic controls were in place for comparison. These were an inverted magnetron gauge (IMG), which monitored the vacuum pressure within the cavity, and a measure of the input RF power coming from the klystrons. The IMG would detect a spike in pressure during a breakdown, whilst the input RF power would drop as the power began to be reflected from the cavity rather than accepted into it. An additional beam-related diagnostic was also used, known as a wall current monitor, this is an electrical pickup along the beam used to monitor beam current. As discussed in the previous section, two measurement methods were tested. The first was the instantaneous readout, which is the same as the method used above for beam loss.

The Cherenkov signal intensity as a function of time was plotted. Measurements could be collected for each pulse of RF power applied to the cavity; in this instance, the pulses had a power of 10 MW and a duration of between 2 μs and 3 μs. With each pulse of RF power, as described above, field emission occurs. The electrons produced and accelerated within the cavity are called dark current. This dark current can be measured using conventional beam diagnostics, such as the wall current monitor. Figure 11 presents the signal from the wall current monitor as a function of time, in comparison to the optical fibre Cherenkov signal. Whereas the wall current monitor is measuring the dark current exiting the cavity, the optical fibre is measuring the loss showers created by the field emission within the cavity. The source of each is the input RF pulse; hence, the characteristics are similar, particularly the timing. Absolute timing was not possible due to a lack of synchronisation between the two systems at the time, but they have been overlapped to emphasise that the relative widths are equal. The Cherenkov signal is also much more intense by a factor of five.

As the width of the RF pulse is varied, in this case between 2 μs and 3 μs, the effect of the field emission was directly visible in the Cherenkov signal. This variation with pulse width is presented in Figure 12. These two results demonstrate an ability to directly monitor the intensity and duration of field emission using a non-invasive optical fibre.

The second measurement method was then assessed, i.e., the integrated signal. This signal is proportional to the charge produced in the field emission process:(9)$$Q=K\frac{1}{R_{L}}\int_{t_{0}}^{t_{1}}\left(V_{meas}-V_{off}\right)dt,$$
where $R_{L}=50\Omega$ is the load, $V_{meas}$ is the measured signal within a time period $\left(t_{0},t_{1}\right)$, $V_{off}$ is the offset value of the detector when no signal is present, and *K* is a calibration constant, whose main contributions encompass the solid collection angle of the fibre, the fibre positioning, and the detection efficiency. In this case, *K* was not calculated, and the relative signal strengths and timings were compared. This integrated signal was collected over the course of RF conditioning. Figure 13 presents the results. The four fibre signals are shown along with the RF input power in orange and the cavity pressure in green. Several details are of note. First, each fibre saw a different signal intensity, but all fibres saw the same signal features, indicating once again the difference that can be produced by different fibre placement and core diameter. Second, as the input RF power was slowly ramped upwards in the RF conditioning process, so too did the integrated signal on all four fibres. This agrees with the earlier instantaneous output results; as the power increases, so does the field emission. Finally, where as Figure 13a presents a period of RF conditioning where the input power was ramped up and no RF breakdowns occurred, Figure 13b shows two RF breakdown events. These are evident where the RF input power is still being applied, yet the IMG pressure reading has spiked. This occurs at a high intensity at 463 min and a low intensity at 471 min. In both instances, all four optical fibres were able to detect a significant increase in Cherenkov radiation generation. Unfortunately, once again, a lack of synchronisation between the new optical fibre system and the existing diagnostics was not possible, so a high resolution comparison of the onset times of these spikes was not possible at this time. There is however a clear corroboration between the existing diagnostics and the optical fibre system. Figure 13b also demonstrates additional features in the Cherenkov signal ahead of the breakdown spike, common amongst all four fibres. Further investigation of these artefacts may lead to novel RF breakdown prediction features.

### 3.3. New System

Following the success of the two measurement campaigns described above, the standout limitation of the optical fibre system was found to be the rise/relaxation time of the SiPM and associated electronics. This was particularly evident in the long tail and electronic ringing found in the beam loss signals, visible in Figure 7. To this end, a new SiPM sensor has been tested alongside new amplification electronics, in collaboration with the beam instrumentation company D-Beam [31]. The new sensor selected was a J-Series SiPM by OnSemi [32] which promised an improved rise/relaxation time as required, but also an improved sensitivity. Alongside this, a new amplifier was procured based on a single current feedback operational amplifier design. This new system promises better performance with easier integration. Unfortunately, beam time on CLARA had finished at the point the new system was ready to be installed. Instead, laboratory measurements using a nanosecond pulsed laser were conducted. As the laser pulse power and duration could be fixed, this would allow a direct comparison with the previous system. Figure 14 presents the results of this comparison. Both of the expected behaviours were observed in the new design. For an input pulse of 5 ns, the new system design is four orders of magnitude quicker and four orders of magnitude more sensitive. As discussed above, Cherenkov radiation is a broadband source, whereas in comparison, the laser used in these measurements has a very narrow line-width; this may lead to some unaccounted differences in efficiency across the full measurable spectrum, but these first results are very promising. This improvement will directly translate to an increase in beam loss location resolution of the order of ∼1 cm, a significant improvement upon the resolution demonstrated in Figure 10. For RF breakdown monitoring, this may open up new avenues of investigation in the artefacts identified ahead of the spikes in Figure 13b.

## 4. Discussion

The work presented here has demonstrated how the detection of Cherenkov radiation using optical fibres has a broad array of applications and uses in the machine protection sector.

The application to beam loss monitoring presented a new upper limit to loss location resolution. The device provides a means of monitoring losses within individual beamline elements; a measurement simply not possible with conventional techniques. The results show how the core diameter of the fibre plays a large role in the signal measured at a single loss location. No discernible resolution difference was found between the 400 μm and 600 μm cores, yet the larger core provided a larger intensity Cherenkov pulse. For this sensor setup, it is more beneficial to use the larger core, then attenuate the signal if required, as per Figure 8. It must be noted however, that with the new detection system this may not be the case; the improved resolution may highlight a discrepancy between the different core diameters and will be investigated further moving forward. A true comparison of the fibre core diameter along the beamline was not possible with this setup, as a consistent fibre layout relative to the beamline could not be achieved for all fibres along the whole beamline.

Application to RF breakdown and RF conditioning proved very successful. Several pre-existing diagnostics were able to provide a strong corroboration of the results produced with the optical fibres. The two different measurement techniques provide an untapped means of further analysis of field emission and RF breakdown onset. It is important to emphasise that these measurements were produced with the exact same system and optical fibre layout as the beam loss measurements. This flexibility would allow operators to easily produce the different machine protection safeguards with less instrumentation than is currently required. That said, future work will look to investigate the specialisation of the device for RF applications. New fibre layouts or specific detector properties may provide deeper insights into machine behaviour than a generalised system could.

The new detector system has demonstrated a superior performance to the existing design in laboratory tests. The goal will now be to replicate the measurements conducted here for beam loss and quantify the improvement in practice. This improved resolution could open opportunities in circular machines, such as storage rings or energy recovery linacs, where particle bunches are tightly packed together. As discussed, it may also draw attention to currently unresolvable features with the Cherenkov signal induced by the RF field emission, features which may provide a predictive capability to the device.

The next stage of this research, aside from the benchmarking of the new detector design, will focus on developing a smart device. This will work to utilise machine learning techniques to probe what can be predicted ahead of time, making this a proactive diagnostic rather than a reactive one. The electronics will be developed to enable this sort of analysis on-board the device, speeding up performance and measurements and opening the door for novel automated machine operation processes. Optimisation of these systems will also require sophisticated simulations of signal generation, capture, and propagation within the optical fibres. This would enable studies into different fibre properties and placements; key for future work. Even without these developments, the disruptive technology presented here, with high resolution, high sensitivity, non-invasive application, and flexibility, will change the way machine protection is conducted within particle accelerators.

## Figures and Tables

**Figure 1 sensors-23-02248-f001:**
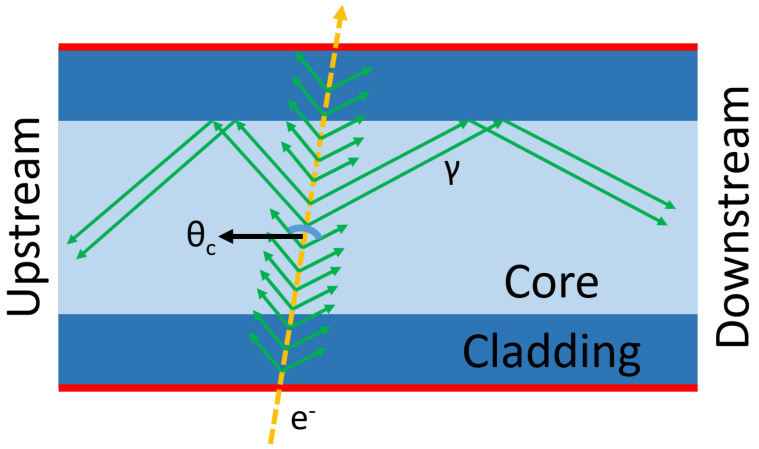
A simplified schematic of the generation, capture, and propagation of Cherenkov radiation in an optical fibre.

**Figure 2 sensors-23-02248-f002:**
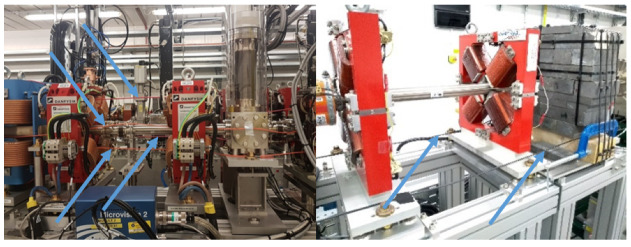
**Left**: optical fibre woven through the electromagnetic elements of a beamline (fibre in orange). **Right**: optical fibre running parallel to a beamline but outside of any machine elements (fibre in black). Fibres are indicated by blue arrows for clarity.

**Figure 3 sensors-23-02248-f003:**
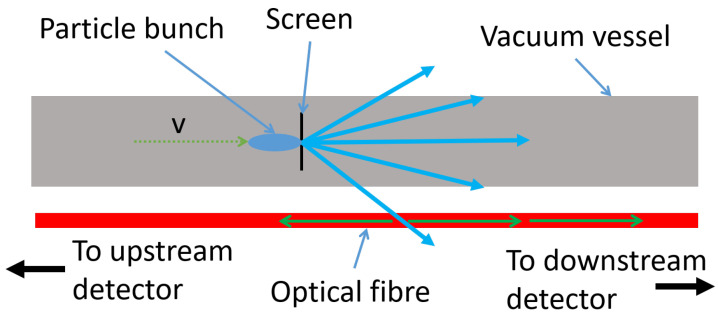
A schematic of a loss event and the location measurement by the optical fibre system. The charged shower is represented by the blue arrows and the subsequent Cherenkov radiation by the green arrows.

**Figure 4 sensors-23-02248-f004:**
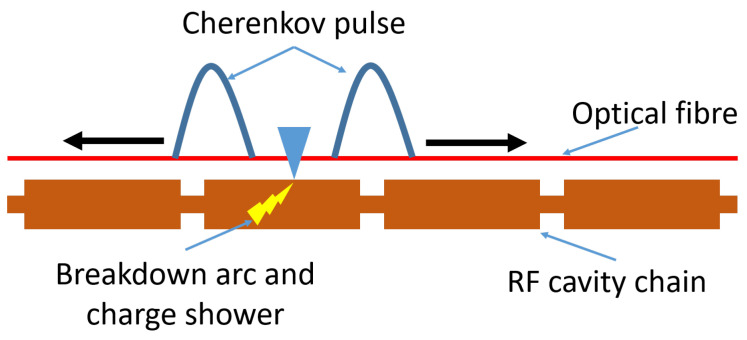
A schematic of a chain of accelerating structures and how the optical fibre system can identify the cavity to cause a breakdown.

**Figure 5 sensors-23-02248-f005:**
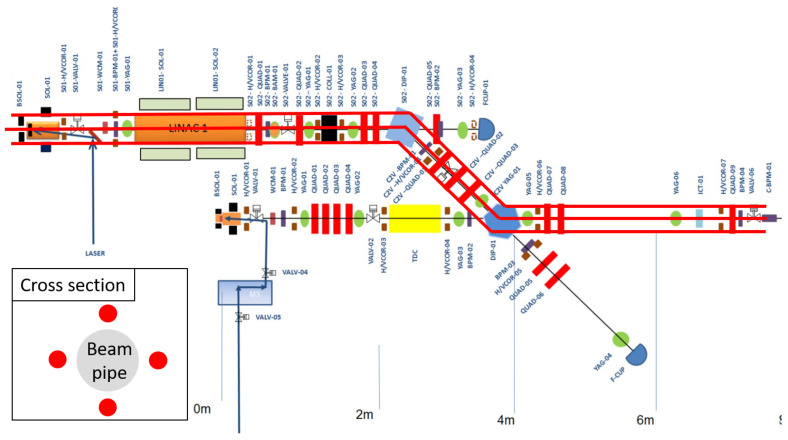
A schematic of the CLARA (STFC, UK) front end, and the approximate locations of the installed optical fibres (red).

**Figure 6 sensors-23-02248-f006:**
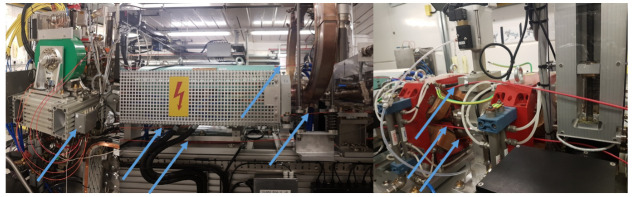
Example images of the optical fibre beam loss monitor installed on CLARA (STFC, UK). Fibres are indicated by blue arrows for clarity.

**Figure 7 sensors-23-02248-f007:**
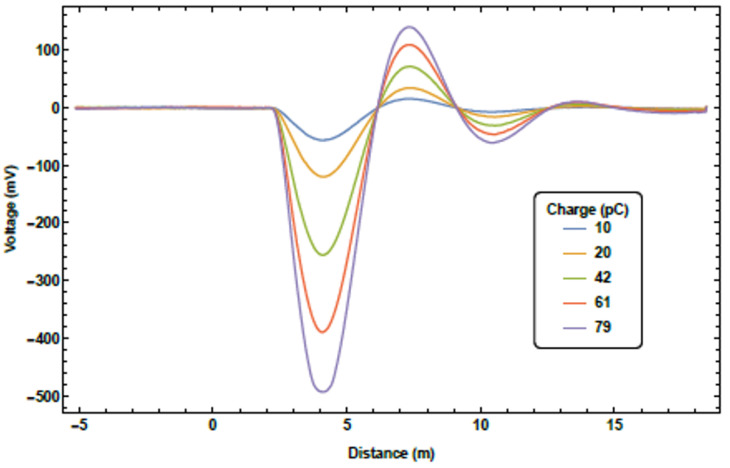
The Cherenkov signal as a result of beam loss for different bunch charges for a single optical fibre.

**Figure 8 sensors-23-02248-f008:**
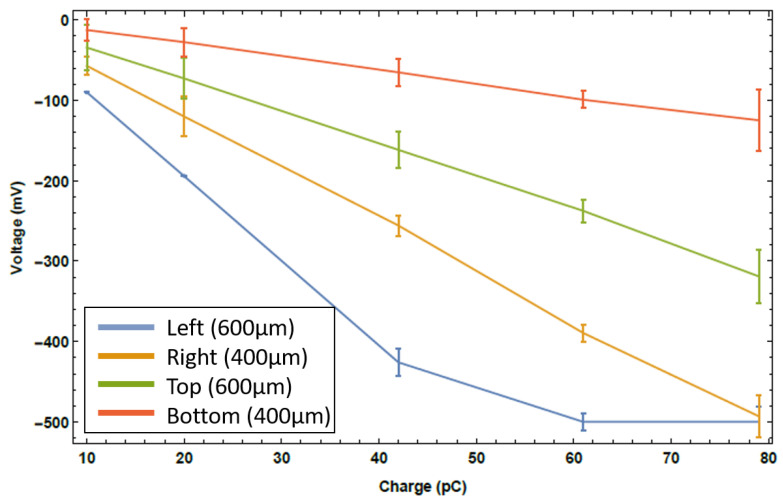
The Cherenkov signal intensity as a result of beam loss as a function of bunch charge for the four optical fibres.

**Figure 9 sensors-23-02248-f009:**
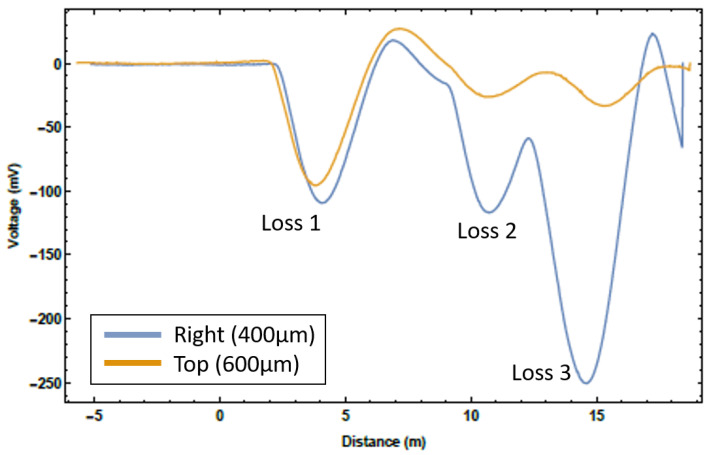
The Cherenkov signal for a three loss location scenario for two of the installed optical fibres. Only signals from two of the fibres are presented for clarity and the sharp peak on the far right is a reflection from the far end of the fibre.

**Figure 10 sensors-23-02248-f010:**
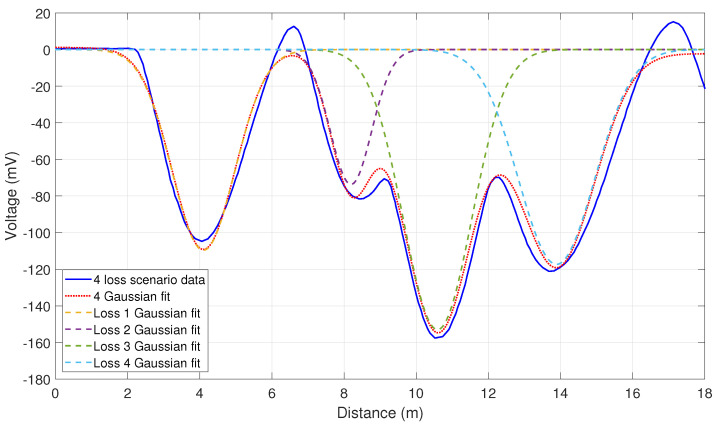
The Cherenkov signal for a four loss location scenario from the right 400 μm core fibre, with a four Gaussian fit and the individual loss Gaussian fits.

**Figure 11 sensors-23-02248-f011:**
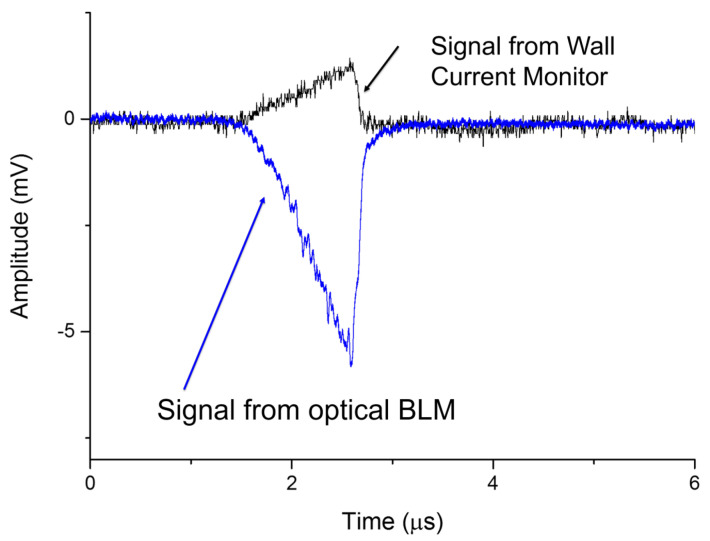
The Cherenkov signal for one fibre as a function of time in comparison with the wall current monitor signal, during a 10 MW 2 μs input pulse of RF power.

**Figure 12 sensors-23-02248-f012:**
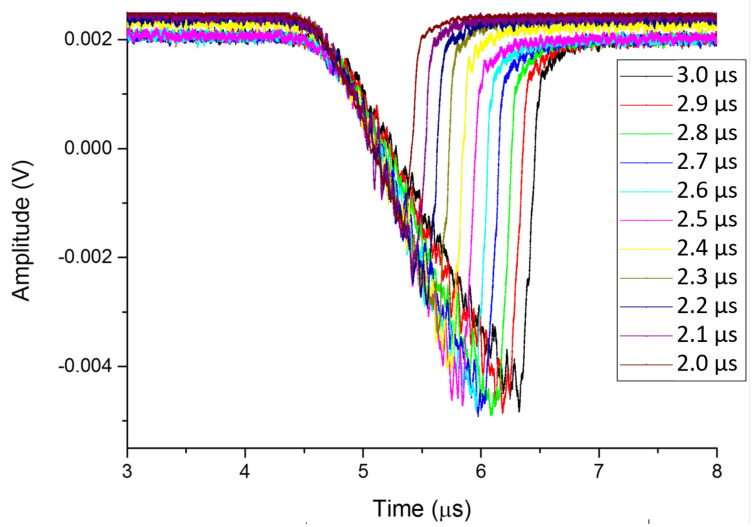
The Cherenkov signal for one fibre as a function of time during a 10 MW input RF pulse with the width varied from 2 μs to 3 μs.

**Figure 13 sensors-23-02248-f013:**
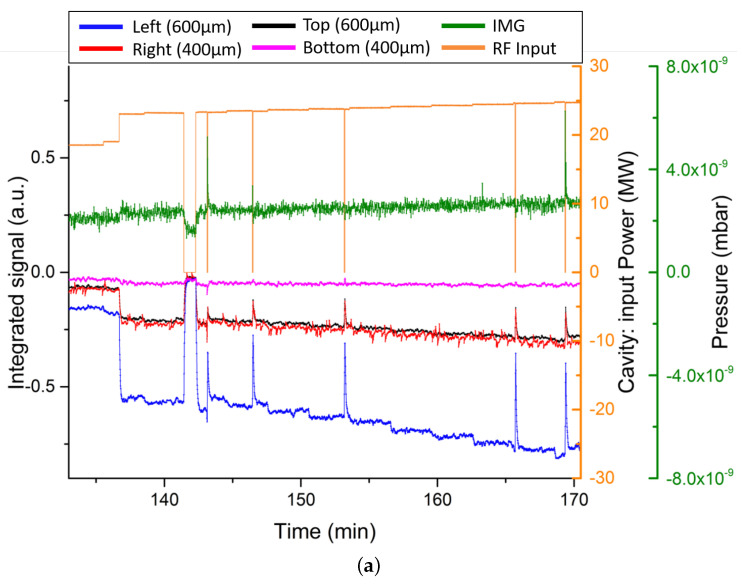
The Cherenkov signals for the four fibres as a function of time during RF conditioning. Furthermore, presented are the readings from the IMG and RF input power during the same time period. (**a**) is for a period of RF input power ramping up with no RF breakdowns and (**b**) includes two RF breakdowns.

**Figure 14 sensors-23-02248-f014:**
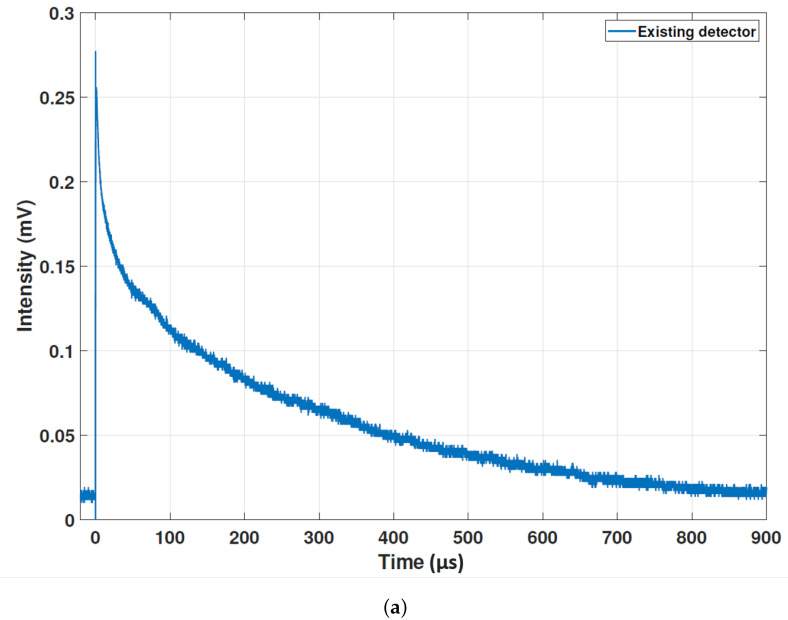
The response to a 5 ns laser pulse in the existing system (**a**) and the new system (**b**).

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
