# Peer review of "Cherenkov Radiation in Optical Fibres as a Versatile Machine Protection System in Particle Accelerators"

_sensors, 2023, doi:10.3390/s23042248_

Round 1

Reviewer 1 Report

This paper described the detection of beam loss position by time-of-flight analysis of Cherenkov light using a silica optical fiber. Although the research is of great significance, the description of the detector and experimental system is insufficient, and the novelty of the results is also problematic. Therefore, unfortunately, I strongly recommend rewriting the paper completely. Below I will describe the problems I found, and I hope you will find them useful.

Section 2.1

The detector configuration is unclear. For example, the model number of the optical fiber, the thickness of the coated tube, the model number of the SiPM, the photosensitive area, etc. The configuration of the TOF readout circuit also seems unclear. It is necessary to provide enough information to enable readers to perform the same experiment.

Section 3.1

Not sure how the beam loss location corresponds to Fig. 5. The scale also does not seem to correspond to Fig. 9, which is ~15 m, and Fig. 5, which is less than 10 m.

It has not been evaluated to what extent the beam loss position in Fig. 9 is accurately detected. Also, how was the 0.5m resolution derived?

Section 3.2

The exact same results and figures in this chapter have already been published in reference 26. The results in this chapter do not appear to be as novel as the original paper.

Section 3.3

The decay time of the existing detector is several hundred microseconds, but this decay time does not seem to be sufficient for TOF analysis at several meters. Is the existing detector different from the systems shown in sections 3.1 and 3.2?

Reviewer 2 Report

The manuscript describes the research and development of a new machine protection system based on the principle that radiation generates Cherenkov light in optical fibers which is innovative, promising and can arouse wide interest. However, the manuscript had some improvements before it could be published. Suggestions for improvement are as follows.

1. In Figure 2, In Figure 2, auxiliary marks of arrows and words should be given.

2. In Figure 8 ,9and 12, “um” should be “μm”, in Figure 11 ,“us” should be “μs”, and in Figure 13 ,“μs” should be “μs”.

3. In line 286 of p.8, the first occurrence of the abbreviation CLARA should state the full name.

4. At the end of Chapter 3, it is necessary to add a comprehensive circuit diagram of light and electricity.

5. It is better to have a conclusion on the determination of the fiber core diameter, that is, the energy deposition of electrons in various fiber cores and even the Cherenkov light yield should be simulated, as well as the measurement of light absorption in the fiber, which is very helpful for designing more reasonable fiber sensors.

Round 2

Reviewer 1 Report

I have received a revised version of the paper. I understand from the author's rebuttal comments that the manuscript is based on non-peer-reviewed conference proceedings and does not use the TOF method. However, it still lacks an explanation of how the positional resolution was assessed. Please correct the following points. I hope you will find them helpful.

Abstract (L12)

In the cover letter, you state that the TOF method is not used in this paper. However, the abstract appears to state that the new system proposed in this study can achieve a position resolution of 0.5 m with the TOF method. Could you please explain this point?

Section 3.1

L345

From the results in Figure 7, you state that the resolution is on the 1 cm scale. However, it doesn't look very likely; it must be a miswriting of the 1 m scale.

L363

How the 0.5 m position resolution was derived is not added in the revised manuscript; loss2 and loss3 in Fig. 9(b) are 2 m apart, indicating that the position resolution is less than 2 m, but it is not clear from this figure whether it is 0.5 m.

Reviewer 2 Report

In the revised version of the manuscript, the author made necessary responses to some suggestions of the reviewer. However, in order to become a scientific and technological document, the quantitative conclusion must be given. The authors have not improved the manuscript according to the reviewer's requirements. Compared with the author's published works in the past, this manuscript has no substantial progress. Therefore, this reviewer does not believe that the manuscript is suitable for publication.. Therefore, the reviewer does not think the manuscript is suitable for publication.

Author Response

The authors would like to thank the reviewer for taking the time to provide additional comments. 

We would like to note that following the first review, only the conclusions was rated as "must be improved", now four sections have been rated as such. This implies the reviewer believes that the first round of revisions made the article worse, but no feedback has been given to this point.

No further feedback points have been given beyond that the previous comments have not been addressed.

There were five previous comments and suggestions.

The first three were implemented.

For the fourth, additions to the text were made and it was explained that all the information requested was already present except the electronic circuit diagram. Reasons for the exclusion of this information were provided but the reviewer has not commented.

The fifth and final comment was asking for a quantitative conclusion on the optimum optical fibre core diameter. As stated in the initial response, and in the article, this was not the goal of the study. Although two fibre diameters were used, a meaningful comparison could not be drawn for reasons stated in the article. The conclusions have been further updated to put further emphasis on this. For our purposes, a larger core provided a larger signal, whereas the benefits of the smaller core in terms of loss location resolution was not found due to the limitation of the existing electronics. Again, the reviewer has not commented on the changes made to the text or on our response.

The Authors therefore feel we addressed all the points raised within the initial review. The second review provided no further actionable comments or suggestions, and rated the article lower on several metrics despite the changes being made as requested where possible. The Reviewer described the work as "innovative, promising and can arouse wide interest" in their first review, but has now recommended it not be published. We therefore strongly disagree with the conclusions of the Reviewer in this instance.